# Scope of Practice and Principles of Care of Naturopathic Medicine in North America: A Commentary

**DOI:** 10.3390/children9010008

**Published:** 2021-12-24

**Authors:** Leslie Solomonian

**Affiliations:** Canadian College of Naturopathic Medicine, Toronto Campus, Toronto, ON M2K 1E2, Canada; lsolomonian@ccnm.edu; Tel.: +1-416-725-6679

**Keywords:** integrative, complementary and alternative, naturopathy, pediatrics, prevention

## Abstract

Naturopathic medicine is a growing profession in North America that provides expertise in complementary and alternative, or integrative care to pediatric patients. It is imperative that healthcare providers have an understanding of the training and scope of other health professionals in order to effectively make decisions regarding research, collaborative clinical care, and policy. Given the prevalence of use of complementary and alternative medicine by children and families in North America, and the growing interest in “integrative” medicine, we aim to offer an overview of naturopathic care for children. This document describes the principles, training, and scope of naturopathic medicine, including health promotion, disease prevention, and illness management. It describes the process by which naturopathic doctors create an integrative healthcare plan for children, evaluate and apply evidence, and integrate ethical issues in practice management, and speaks to the role naturopathic doctors have regarding advocacy for community and planetary health as it relates to pediatrics.

## 1. Introduction

The 2007 National Health Interview Survey (NHIS) revealed that approximately one in nine children (11.8%) used complementary and alternative medicine (CAM) in the previous 12 months [1]. Some children are significantly more likely to use CAM (also referred to as integrative medicine), including children with mental health issues [2], pain conditions [3], respiratory concerns [4], cancer [5], and neurological concerns [6]. Families often do not consult with a healthcare provider about the use of integrative medicine, and may not disclose its use to their medical doctor [4]. In the 2012 NHIS survey, only 18.4% of CAM usage had been recommended by a medical doctor [2]. It is critical that parents and guardians are guided by healthcare providers thoroughly trained in integrative approaches to pediatric care. Additionally, the establishment of healthy habits and the promotion of optimal lifestyle are core components of pediatric practice, influencing individual and community health through the lifespan.

Although some medical schools incorporate integrative approaches and lifestyle medicine into their curricula [7,8], naturopathic doctors (NDs) are the only professionals that are trained primarily in these approaches at a doctoral level. This ideally positions them as experts in these paradigms, to ensure safety and effectiveness for families when working independently or in collaboration with other providers. 

This commentary provides information about current naturopathic training and scope in North America, and articulates guiding principles of naturopathic medicine in the care of children and adolescents. The aim is to better equip other healthcare providers to make decisions regarding collaborative clinical care. It reflects what is taught within the accredited naturopathic medical institutions, and what is promoted by the Pediatric Association of Naturopathic Physicians’ (PedANP) through member resources and continuing education. 

This document was developed on behalf of the “Principles of Care Committee,” and approved by the board of directors. The PedANP is an independent affiliate of the American Association of Naturopathic Physicians, which represents and advocates for advancement of the naturopathic profession in the United States. The mission of the PedANP is to expand and disseminate knowledge about best practices in naturopathic medical care for the pediatric population, to elevate the standard of pediatric care and education in naturopathic medical programs and the profession as a whole. Elected members of the board of the PedANP are experts in the field of naturopathic pediatrics who practice in a range of clinical and academic settings. 

## 2. Paradigm, Philosophy, Training and Regulation of Naturopathic Practice

Medical pluralism is an important part of the history of the naturopathic profession [9]. The roots of modern naturopathic training and practice lie in the integration of diverse approaches to health from a variety of doctrines. Some practitioners lean toward more traditional approaches (such as acupuncture or traditional herbal medicine), while others favor methods that have well-understood biomedical mechanisms and a rigorous evidence base. Many work with a combination of the two. Some NDs emphasize lifestyle strategies, while others rely more on the use of natural health products and other CAM approaches. 

The principles of naturopathic medicine act as a touchstone for NDs (Figure 1). As all healthcare providers are, NDs are guided by the fundamental principle of nonmaleficence; five additional principles inform naturopathic care and decision-making, all of which are emphasized throughout training [10,11]. The Therapeutic Order was designed as a framework for holistic care to which these principles are applied (Figure 1). While a critical part of an ND’s role is identifying when more forceful methods and/or referral may be necessary, the principles of the profession remind clinicians to approach families and children holistically, individually, and with the least force possible in order to achieve the therapeutic goal. 

There are an estimated 5000 licensed NDs practicing in North America. Regulated jurisdictions include 5 provinces, 22 states, the District of Columbia, Puerto Rico and the US Virgin Islands (please visit www.naturopathic.org, accessed on 21 December 2021 or www.cand.ca, accessed on 21 December 2021, for details). The broadest scopes (such as is that in the state of Arizona) include pharmaceutical prescribing, vaccination administration, intravenous nutrient therapy, acupuncture, and minor surgery. Licensed naturopathic doctors (NDs) complete an accredited (www.cnme.org, accessed on 21 December 2021) four year post-graduate medical education which blends conventional biomedical training with traditional modalities, including clinical nutrition, mind-body medicine, lifestyle counselling, botanical medicine, hydrotherapy, and hands-on manual therapies. Some institutions offer additional training in other domains such as Traditional Chinese Medicine and acupuncture, midwifery, and public health. There are currently eight accredited programs for naturopathic medical education in North America.

All licensed or regulated jurisdictions require successful completion of two sets of standardized board exams (https://www.nabne.org/about/, accessed on 21 December 2021), as well as relevant local regulatory exams. All NDs are trained as generalists. Residency training is currently not a requirement for NDs, although opportunities exist, including those with a focus on pediatrics. Specialization is not recognized in most jurisdictions. However, the American Board of Naturopathic Pediatrics (an independent arm of the PedANP) offers a fellowship certification process (https://www.pedanp.org/board-certification, accessed on 21 December 2021), which is recognized as evidence of board-certification in some regulated jurisdictions of practice.

## 3. Naturopathic Pediatric Practice

Naturopathic doctors act in pediatric care in both primary and consultative roles. NDs are trained to promote and monitor healthy growth and development, educate families on preventative strategies, address risk factors for disease, and manage childhood illnesses. As all primary care providers do, NDs may play a number of roles in the care of children: Wellness promotionPrevention of diseaseSurveillance and screening for developmental delay or diseaseManagement of acute and chronic illnessCoordination of careAdvocacy for a healthy community and environment

### 3.1. Wellness Promotion and Prevention of Disease

Working with children and families is the epitome of prevention and health promotion. When a healthy childhood is fostered, there is tremendous potential for lifelong individual and community health. Achieving these purposes requires an appreciation of the determinants of health, and normal growth and development. Progress in hygiene, nutrition, and medical innovation in the past century has shifted the focus of North American pediatric practice from managing infection and malnutrition to preventing chronic conditions that have roots in early life [12,13,14,15]. There is a plethora of evidence highlighting the importance of fostering good lifestyle practices, such as optimal diet, physical movement, sufficient sleep, minimal exposure to environmental toxins, and stress mitigation, and the social and ecological determinants influencing these. Practice models generally ensure that encounter time is adequate to engage with families regarding the determinants of health most relevant to the child, their family, and their broader community, and to offer strategies to promote wellness in both the short and long term (Figure 2) [16]. 

Naturopathic students—like all healthcare trainees—are also trained to seek opportunities for surveillance and anticipatory guidance at every visit, and regular wellness-focused visits are often encouraged to more THOROUGHLY engage in health promotion and the sequential monitoring of healthy growth and development [16]. Well-child checks are typically advised to be consistent with recommended immunization schedules. Although most guidelines recommend annual well-child visits after 2.5 years of age, thorough health promotion may be more effective if done semi-annually, quarterly, or more frequently if it appears to be necessary to address all concerns [22,23,24,25]. Naturopathic doctors are trained to utilize evidence-informed guidelines for screening, well-child promotion, and primary prevention [26] to ensure key determinants of health are addressed. As they do for all primary healthcare providers, well-child visits consist of a thorough history, a comprehensive physical exam, any indicated objective screening measures, relevant labs, and anticipatory guidance, depending on scope of practice. Naturopathic doctors in some jurisdictions can and do offer the full scope of recommended childhood immunizations. NDs in other jurisdictions cannot provide this healthcare service, but may engage in discussions of immunization to encourage informed consent. 

NDs typically engage in detailed lifestyle education and anticipatory guidance specific to the age, development, and health status of the patient with a strong emphasis on informed consent and empowered decision-making. Families are typically asked to attend a well-child check with a recent 7-day diet/sleep/activity/screen use diary [27].

### 3.2. Management of Disease—Creating an Integrative Pediatric Plan 

When a child presents with a health concern, either acute or chronic, the ND is trained to seek to explore the cause(s) of the concern, both proximal and distal. A core aspect of the naturopathic paradigm is to treat the individual in a holistic manner; as such, the perceived cause, and thus proposed solutions, may be different from child to child. In jurisdictions where NDs are fully integrated into the healthcare system (including prescriptive authority for pharmaceutical medications, laboratory and imaging access, ability to refer to specialists, and coverage with health insurance programs), NDs are equipped to engage in primary care, including referral to or coordination of care with other pediatric specialists when indicated. In other jurisdictions, it is ideal for families to also establish a relationship with a medical doctor in order to facilitate access to the full health system, and are encouraged by their ND to do so. 

Naturopathic doctors are taught to draw on conventional standards of care and algorithms to guide diagnostic reasoning and management options, while factoring in frameworks from other medical paradigms (for example, consideration of a Traditional Chinese Medicine diagnosis). Depending on the nature of the illness, the values of the family, and the experience and scope of the clinician, a variety of therapeutic options may be considered [28]. This may include the use of botanical medicines (either topical or oral), other natural health products such as nutritional supplements or probiotics, or hydrotherapy (the therapeutic application of hot and cold water to manipulate circulation). NDs with the training and authority to do so may prescribe or administer pharmaceutical medications; or perform acupuncture, acupressure, or manual therapies (such as soft tissue or joint manipulation). Those without such authority are trained to appropriately refer to other providers when indicated. Management plans typically include recommendations for optimizing lifestyle factors such as diet, sleep habits, physical activity, stress mitigation and spending time in nature. 

Regardless of the jurisdictionally defined age of consent for medical intervention [29,30,31], engaging children in decision-making and obtaining assent from them directly is a key part of increasing motivation to follow through with a given plan, which is of added importance when recommending the behavior modification or lifestyle changes common in naturopathic practice. The feasibility of all recommendations are ideally thoroughly explored to minimize conflict within the family environment and maximize success. It is unrealistic, for example, to significantly modify the diet of a child without considering other members of the family. Lifestyle recommendations specific to a particular child may in fact positively benefit the health of the entire family. The child should be included in planning as much as is developmentally possible to establish trust and rapport. Ideally, the child will be empowered by the plan, and take an appropriate level of responsibility in executing it. This requires some level of creativity and negotiation of choices. Practitioners attempt to engage strategies to educate, inspire, and motivate the family, making recommendations that are relevant to the unique determinants of that child’s health.

### 3.3. Evidence-Informed Naturopathic Practice and Informed Consent

In close collaboration with the family, NDs develop a plan that takes into account a holistic evaluation of the patient; available evidence for safety, efficacy, and interactions of conventional and complementary management strategies; and the family’s values and preferences in order to develop an individualized plan. Given the potential range of approaches that an ND may recommend, it is the duty of the clinician, as it is for all healthcare providers, to present these options to the child and their family in a way that enables true informed consent. This conversation requires discussion of the risks and benefits of each option, including what would be considered the conventional standard of care, and the natural progression if left unaddressed. 

Naturopathic doctors consider each intervention in terms of the available evidence for its safety and efficacy for a particular scenario (Figure 3) [32]. A dearth of high-quality evidence is a challenge that affects both naturopathic and pediatric care independently, made worse by the combination. Application to children may need to be extrapolated from literature relevant to adults, and from pediatric studies focusing on safe and effective management of other related conditions if available. During a review of the available literature, clinicians are trained to look for reported adverse events and evidence of tolerability. Since physiologically, metabolically, and behaviorally, children are not just small adults [33], caution should be taken to minimize the risk of harm. NDs are trained to adjust dosing of natural health products to the age and weight of the child [34]. Leaning toward less invasive or toxic interventions in general minimizes risk. 

Potential risks are not only biomedical, but also financial, psychosocial (e.g., expensive, complicated protocols with little benefit), or harms of omission (allowing a child to suffer needlessly through not appropriately referring. NDs are well trained to anticipate potential interactions between natural therapies and conventional interventions the child may be using. 

Meta-analyses and randomized controlled trials do not always reflect the holism of naturopathic and client-centered care. True evidence-informed practice relies on the available literature, but also on the clinical experience of the practitioner, and the values and preferences of the patient or family, which includes the choice to seek the support of an ND [35]. All three must be integrated into the process of obtaining informed consent which requires a thorough discussion of all perceived risks and benefits of an approach and viable alternatives [36]. Ultimately, each intervention is an n = 1 trial—true for all healthcare, and a core tenet of naturopathic practice—and should be navigated as such with the family (see Figure 4). NDs are trained to layer these thought processes on top of a foundation informed by naturopathic principles, which may ultimately influence the interpretation of evidence and/or preference by the patient or family. 

### 3.4. Ethical Issues in Practice Management

Naturopathic doctors are expected to manage and disclose any potential conflict of interest, including financial gain from the sale of natural health products. NDs are also mandatory reporters of suspected child abuse and neglect in Canada and United States and receive training on managing these challenging scenarios [37]. Naturopathic doctors are beholden to all standard limits of confidentiality within the jurisdiction of practice. 

### 3.5. Advocacy for a Healthy Community and Environment

Naturopathic doctors are trained to strive for holistic care which integrates not only all aspects of an individual’s health, but also its determinants, including the local and global environment. Poverty and socioeconomic factors are key determinants of health, affecting nutrition, stress, toxic environmental exposure, access to the outdoors, access to education, and an increased chance of risky behaviour [38,39,40,41]. Black, indigenous, and other people of colour are disproportionately affected. Human-caused ecological degradation of the natural world has a multitude of detrimental effects on human health [42]. Children are particularly vulnerable, due to the immaturity and plasticity of tissues and organ systems, as well as their disproportionately greater uptake due to normal physiology and behaviour [43,44]. They will also be most impacted in the future by widespread ecological and societal breakdown due to climate devastation [45]. Vulnerable and marginalized populations are the first and most significantly affected by environmental impacts [46,47].

While these systemic factors are significant determinants of pediatric health, they also often place naturopathic care out of reach to vulnerable youth. In an effort to contribute to the recognized necessity of a prevention-oriented, low-carbon, sustainable, and equitable health system, some NDs aim to improve access to their services for those who are vulnerablized; deliberately provide education on the reciprocal connections between planetary, community and individual; make “nature prescriptions”; establish personal and professional habits to minimize their personal environmental footprint; and engage in advocacy for social and climate justice action. 

## 4. Referral to and Collaboration with a Naturopathic Doctor

Given the heavy emphasis on the optimization of health and prevention of disease, arguably all children would benefit from consultation with an ND, whether or not there is a particular health concern. While other front-line healthcare providers value these priorities, they may be limited in their capacity to implement them due to systemic barriers such as training and billing practices [48,49]. The “why” and “how” of lifestyle medicine is a core focus of naturopathic training, and NDs spend a notable amount of time engaged in lifestyle counselling and strategies of behaviour change. Collaboration with an ND can ensure families receive pragmatic support in terms of anticipatory guidance and holistic lifestyle counselling. This allows others in the medical community to focus on their areas of expertise, while NDs can apply their unique lens to enhance the capacity of the child to remain well and heal efficiently from the inevitable health concerns of childhood. 

A cross-sectional analysis of naturopathic practitioners surveyed indicated that the most common reasons for pediatric naturopathic visits were health supervision, acute infection, and mental health concerns [50]. Interest in and evidence for the use of CAM approaches in general are greater in certain populations, as mentioned earlier. As the only profession for which a doctorate is accredited for CAM, NDs are an excellent member of the healthcare team for families who are particularly interested in this approach to health. Because NDs are trained in both the biomedical model and other paradigms of healthcare, they are well-positioned to bridge any gaps through deeper conversations around the interpretation of evidence for CAM approaches, and strategies for integrating the best of both worlds. 

If seeking to refer to an ND in your community, it is important to ensure the practitioner is appropriately trained and credentialed. A suggested due diligence checklist may include:If your jurisdiction is regulated, is this individual registered? Visit https://naturopathic.org/page/RegulatedStates (accessed on 21 December 2021) and https://www.cand.ca/common-questions-education-and-regulation/ (accessed on 21 December 2021 for more information.If yours is not a regulated jurisdiction, it is advised to request evidence of successful completion of training from an accredited naturopathic institution and successful completion of board exams.Many naturopathic doctors who practice in unregulated jurisdictions wisely maintain their status in a regulated state or province to provide documentation of good standing. This requires regular engagement in continuing education and maintenance of malpractice insurance.The PedANP maintains a list of professional members. However, membership only suggests an interest in pediatrics, not a specialization. The Fellowship of the American Board of Naturopathic Pediatrics (FABNP) designation indicates that the individual has met standards of certification, but this is not an accreditation of exclusion. Any naturopathic doctor may see children.

## 5. Conclusions

CAM use in pediatric medicine is becoming increasingly popular in North America. It is vital that parents and caregivers who seek integrative methods for managing their children’s health consult health professionals who are properly trained in natural, holistic, and integrative approaches to pediatric care. Naturopathic doctors licensed in North American jurisdictions are uniquely trained in this paradigm, positioning them well to be integral members of care teams for children. Other healthcare providers are encouraged to become familiar with the regulation of NDs in their region of practice, and refer patients who would benefit from competent, evidence-informed integrative care. Through a unique emphasis on holistic, individualized care, prevention, and health optimization, co-management with a ND creates space for other healthcare providers to excel at the roles for which they are appropriately trained.

## Figures and Tables

**Figure 1 children-09-00008-f001:**
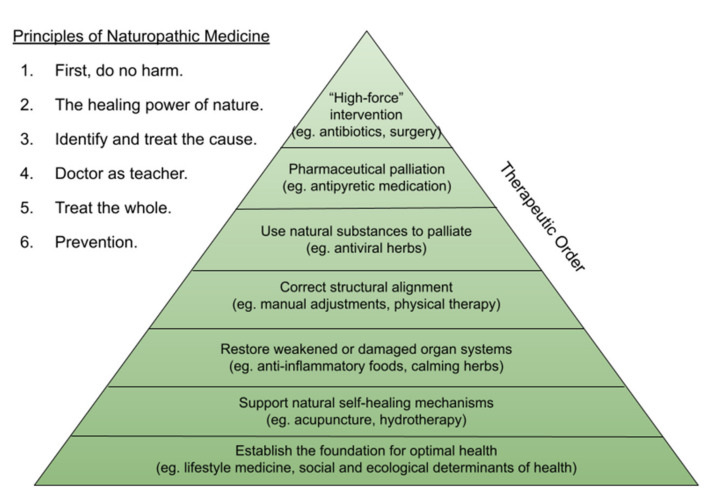
The six principles of naturopathic medicine act as a set of guidelines to inform decision making for holistic care. Therapeutic order is a framework to which these principles are applied.

**Figure 2 children-09-00008-f002:**
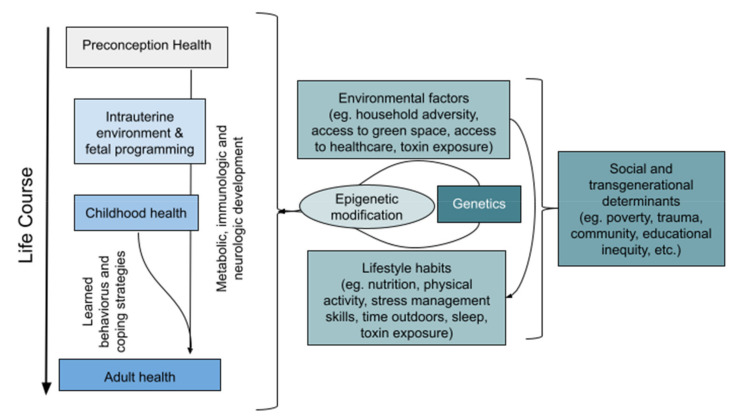
Many factors are established early in life that affect adult health outcomes. Some factors are modifiable, such as lifestyle habits and social behaviours. Others are pre-determined (but may be modifiable) such as preconception health, genetics, and transgenerational behaviour patterns. (courtesy of author, informed by references [17,18,19,20,21]).

**Figure 3 children-09-00008-f003:**
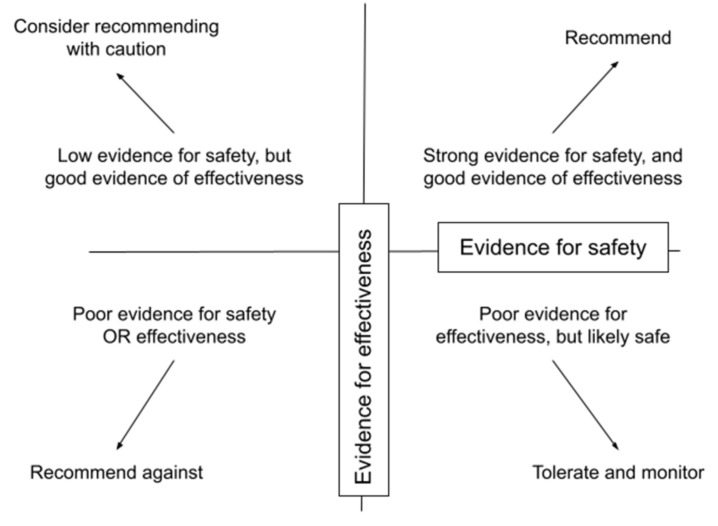
Naturopathic doctors consider each intervention in terms of the available evidence for its safety and efficacy in order to help the patient make informed choices for a particular scenario.

**Figure 4 children-09-00008-f004:**
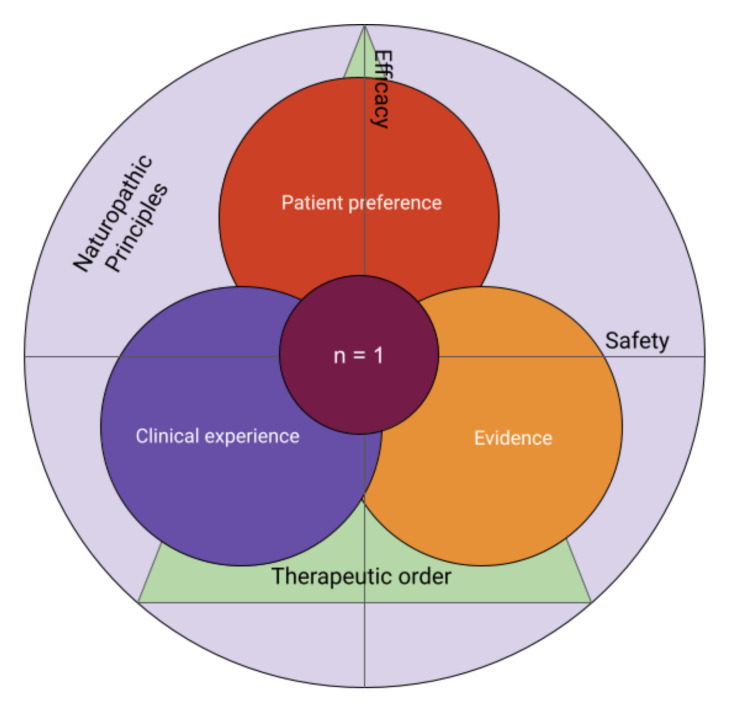
Evidence informed practice considers the available evidence for safety and efficacy, reflective clinical experience, and the values and preferences of the patient or family in order to inform consent, ultimately leading to an “n-of-1” trial.

## Data Availability

Not applicable.

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
