# Peer review of "Scope of Practice and Principles of Care of Naturopathic Medicine in North America: A Commentary"

_children, 2021, doi:10.3390/children9010008_

Round 1

Reviewer 1 Report

This discussion paper outlines the scope and practice of naturopathic medicine as it pertains to pediatric care and is an important contribution to public health. The ability to understand how naturopathic medicine in the context of pediatric care is practiced can help other health care practitioners know when and why a referral might be relevant to members of this profession and also to better understand the level of accountability that NDs can offer in the care of mutual patients. Overall, the author is to be commended on the writing, structure and thought put into this summary approach taken by NDs in the context of pediatric care.

Most of my comments are relatively minor in terms of suggesting some improvement to the paper overall. 

Page 2 - Section 2:

With the first use of the term naturopathic doctor, the shorthand acronym ND should be introduced rather than in section 3 as it is currently. From the earliest point on, perhaps the shorthand ND could be used more frequently throughout.

The provision of Figure 1 with the therapeutic order alongside the principals of care is an excellent contribution to the paper.

Use of what is done vs what ‘should’ be done something to consider. It seems through much of the discussion of how NDs should treat their pediatric patients optimally there is an emphasis on what is done rather than what ought to be done. It might be better in some cases to emphasize what should be done in the idealized practice rather than stating what is done in all cases. For example the following sentence as part of the Prevention and Wellness section states the following:

Opportunities for surveillance and anticipatory guidance are sought at every visit, and regular wellness-focused visits are encouraged to more thoroughly engage in health promotion and sequential monitoring of healthy growth and development.

Consider reframing with instead of “are” to be “should” which indicates the idealized practice rather than what is an absolute in terms of delivery.

I won’t point out the other instances of this which are frequent, and should be considered as a whole in any revision.

Figure 2 is an excellent contribution to the paper.

Page 6 – paragraph 2: When discussing the need to get consent may be useful to note the need to get ‘assent’ from a child even though they may not legally be required to given consent. Assent indicates their willingness and non-resistance to a particular intervention or therapy.

Figure 4 – I like the idea of this figure very much and understand the basic elements, but would consider revisiting and revising how the information is presented to make it clearer if possible. I understand what is being shown, but the triangle overlaid on top of the cross for instance is confusing and may reiterate the triangle of hierarchy where it’s trying to identify the triad of patient preference, clinical experience and evidence. I would first of all remove the triangle which shows the ‘hierarchy of evidence’ and instead of a triangle linear lines linking the three elements, perhaps to have circular links instead in the consideration of true ‘evidence based medicine’. Also, if you are going to include naturopathic principles in the image this should probably be placed next to clinical experience component not the patient preference. May be good to include the component on the therapeutic order as well here.

One additional point that could be brought forward more clearly is the organization within the naturopathic profession that is most focused on pediatric care. If this group provides additional training and education in this area it would be good to make this explicit. The inclusion of a note of review by the PedANP board is worth elaborating on and what role this organization has within the profession.

Lastly, it might be good to consider in what situations it would be ideal for a patient to be referred to an ND. Is it in all scenarios that care can be considered and/or are there most relevant situations where a referral or consultation with an ND most appropriate?

Author Response

Reviewer comments

Author response

This discussion paper outlines the scope and practice of naturopathic medicine as it pertains to pediatric care and is an important contribution to public health. The ability to understand how naturopathic medicine in the context of pediatric care is practiced can help other health care practitioners know when and why a referral might be relevant to members of this profession and also to better understand the level of accountability that NDs can offer in the care of mutual patients. Overall, the author is to be commended on the writing, structure and thought put into this summary approach taken by NDs in the context of pediatric care.

The provision of Figure 1 with the therapeutic order alongside the principals of care is an excellent contribution to the paper.

Figure 2 is an excellent contribution to the paper.

Thank you for your constructive feedback, so clearly demonstrating your understanding of our intention of this manuscript. 

Page 2 - Section 2:

With the first use of the term naturopathic doctor, the shorthand acronym ND should be introduced rather than in section 3 as it is currently. From the earliest point on, perhaps the shorthand ND could be used more frequently throughout.

Thank you for identifying that this is better offered earlier, and used more frequently.  The revised manuscript reflects these changes.

Use of what is done vs what ‘should’ be done something to consider. It seems through much of the discussion of how NDs should treat their pediatric patients optimally there is an emphasis on what is done rather than what ought to be done. It might be better in some cases to emphasize what should be done in the idealized practice rather than stating what is done in all cases. For example the following sentence as part of the Prevention and Wellness section states the following:

Opportunities for surveillance and anticipatory guidance are sought at every visit, and regular wellness-focused visits are encouraged to more thoroughly engage in health promotion and sequential monitoring of healthy growth and development.

Consider reframing with instead of “are” to be “should” which indicates the idealized practice rather than what is an absolute in terms of delivery.  I won’t point out the other instances of this which are frequent, and should be considered as a whole in any revision.

This is an excellent distinction.  Since the author and the advisory committee are all engaged in teaching and delivering “idealized” pediatric care, we hope that most NDs do, in fact, practice this way.  

The manuscript has been revised to clarify this as an overarching principle (please see page 2), and opportunities to adjust this implication have been taken throughout.

Page 6 – paragraph 2: When discussing the need to get consent may be useful to note the need to get ‘assent’ from a child even though they may not legally be required to given consent. Assent indicates their willingness and non-resistance to a particular intervention or therapy.

Excellent distinction.  The manuscript has been revised to reflect this change.

Figure 4 – I like the idea of this figure very much and understand the basic elements, but would consider revisiting and revising how the information is presented to make it clearer if possible. I understand what is being shown, but the triangle overlaid on top of the cross for instance is confusing and may reiterate the triangle of hierarchy where it’s trying to identify the triad of patient preference, clinical experience and evidence. I would first of all remove the triangle which shows the ‘hierarchy of evidence’ and instead of a triangle linear lines linking the three elements, perhaps to have circular links instead in the consideration of true ‘evidence based medicine’. Also, if you are going to include naturopathic principles in the image this should probably be placed next to clinical experience component not the patient preference. May be good to include the component on the therapeutic order as well here.

Thank you for this feedback from fresh eyes and brain. I have adapted this image in an attempt to better reflect its intention.

With respect to the suggestion to place naturopathic principles (and possibly therapeutic order) next to clinical experience, it is my observation that most patients choose naturopathic doctors because of these lenses.  It is a transparent part of the paradigm.  In a similar way, naturopathic doctors are trained to look at evidence using these lenses as well; not only reducing a study to minimal variables, but questioning the relevance of evidence within a holistic paradigm. I have added some text in the manuscript to capture this (page 7).

One additional point that could be brought forward more clearly is the organization within the naturopathic profession that is most focused on pediatric care. If this group provides additional training and education in this area it would be good to make this explicit. The inclusion of a note of review by the PedANP board is worth elaborating on and what role this organization has within the profession.

Thank you for this suggestion.  I have attempted to elaborate on this piece (page 2 and 3).

Lastly, it might be good to consider in what situations it would be ideal for a patient to be referred to an ND. Is it in all scenarios that care can be considered and/or are there most relevant situations where a referral or consultation with an ND most appropriate?

Thank you for this prompt.  I have added a section to address this suggestion.

Reviewer 2 Report

Authors must first adapt the manuscript according to the journal's standards for reviews: "These provide concise and accurate updates on the latest advances in a given research area. Systematic reviews should follow PRISMA guidelines".

Author Response

 It appears as though this reviewer assessed this manuscript as though it was intended as a systematic review.  Indeed, this was intended as a commentary or position paper, as opposed to a review of the literature.  Unfortunately, sources to inform such a review do not exist.  It was shaped by experienced practitioners who are leaders of the regulated and accredited naturopathic medical profession in Canada and the US, and as such, reflects how the practice of pediatric naturopathic medicine is taught and encouraged within the profession.  We were unable to identify a submission type for a commentary or position paper.  We leave it to the editors to ascertain if this manuscript is suitable for this journal.

Reviewer 3 Report

This manuscript provides a summary of naturopathic approaches to paediatric care. It is generally well written however it is unclear what its purpose is. It is clearly not a literature review as large portions are unreferenced and read as opinion rather than an unbiased critique of the current research literature. I’m also not sure of who the audience of this work is. It appears that this the work of committee rather than a single individual which is not reflected in the author information. This presents an ethical issue as all contributing authors should be listed. In many respects this appears to be a position paper and I suggest that given the statements at the end of paragraph 2 of the Introduction this would be better presented as a position paper/editorial with the authors the PedANP rather than a single individual. Given the above it is hard to provide specific comments; if this is a literature review then it needs considerably more work to situate it in the literature and give it a more objective feel. If, however it is intended as a position paper/editorial from the PedANP then I have fewer comments and the burden of evidence and objectivity is lessened.

Author Response

Indeed, this was intended as a commentary or position paper, as opposed to a review of the literature.  Unfortunately, sources to inform such a review do not exist.  It was shaped by experienced practitioners who are leaders of the regulated and accredited naturopathic medical profession in Canada and the US, and as such, reflects how the practice of pediatric naturopathic medicine is taught and encouraged within the profession.  We were unable to identify a submission type for a commentary or position paper.  

As the author of note, I can affirm that I was responsible for the crafting of this manuscript on behalf of the PedANP, the board of which reviewed, advised and approved the final manuscript.  We would be very pleased to make that clear within the guidelines of the journal, and are submitting a change of authorship form. 

Round 2

Reviewer 2 Report

Due the relevance of this issue, in my opinion, this manuscript can be published such us "editorial" or "opinion piece".

Author Response

"Due the relevance of this issue, in my opinion, this manuscript can be published such us "editorial" or "opinion piece"."

I appreciate this feedback.  I'm glad the intention of the manuscript is more clear than it had previously been.  It was written as a means of providing information to other professionals who care for children about the role of naturopathic doctors in the circle of care.  I defer to the editorial staff as to the best way to frame it in the context of the publication.  Given it is written on behalf of an association of naturopathic doctors who care for children, it could be seen as an editorial, commentary, or position statement. 

Reviewer 3 Report

The revision by the author has somewhat addressed the previous comments how the main issue remains is that this is submitted as a research review and it does not meet the essential requirements for a review. There is little to no engagement with the literature and no critical analysis or insight rather this is an editorial or opinion piece. I will leave it to the editor as to whether there is capacity within the framework of the journal to publish a piece such as this.

The new text (section 4) has the statement “Given the heavy emphasis on optimization of health as opposed to only management of illness, all children would benefit from consultation with an ND, whether or not there is a particular health concern.” yet there is no supporting evidence to validate this claim other than the view of NDs that this should happen. Similarly, the comment “Because medical doctors, doctors of osteopathy, and nurse practitioners are often limited in their opportunities to offer health promotion guidance, …” is unfounded and unnecessary. I find many of the statements in this section about other health professions inflammatory and presented in an unbalanced way.

Author Response

"The revision by the author has somewhat addressed the previous comments how the main issue remains is that this is submitted as a research review and it does not meet the essential requirements for a review. There is little to no engagement with the literature and no critical analysis or insight rather this is an editorial or opinion piece. I will leave it to the editor as to whether there is capacity within the framework of the journal to publish a piece such as this."

Thank you for reinforcing this.  We/I agree that this is not a research review, nor was it written to be so.  It was submitted with the intention of being an editorial, commentary, or position piece to provide information to other healthcare providers about the naturopathic care of children.  I too defer to the editorial team to determine if it is appropriate within this publication.  

"The new text (section 4) has the statement “Given the heavy emphasis on optimization of health as opposed to only management of illness, all children would benefit from consultation with an ND, whether or not there is a particular health concern.” yet there is no supporting evidence to validate this claim other than the view of NDs that this should happen. Similarly, the comment “Because medical doctors, doctors of osteopathy, and nurse practitioners are often limited in their opportunities to offer health promotion guidance, …” is unfounded and unnecessary. I find many of the statements in this section about other health professions inflammatory and presented in an unbalanced way."

I am very grateful for this feedback!  In no way was it intended to be inflammatory or dismissive.  Another reviewer suggested including a statement to guide other healthcare providers as to the circumstances that may benefit from referral to or co-management with an ND.  Thus the new section.  It is challenging to provide supportive evidence to something that is based on principles of practice, such as the emphasis on health promotion and lifestyle medicine offered by NDs.  Contrary to it being the opinion of the author, holistic, individualized, preventative care is the foundation of naturopathic practice.  The therapeutic order emphasizes that in addition to managing illness, optimizing conditions for health is a priority.  As such, what naturopathic doctors are trained to is to treat the person and not the illness.  Thus the statement that all children would benefit from consultation with an ND, to assess and promote healthy habits such as optimal nutrition, sleep habits, physical activity, stress management, while factoring in the unique determinants (including social and ecological) of these behaviours.  The suggestion that other providers are limited in their opportunities to do so is in reference to healthcare systems and billing structures that interfere with the time and training other healthcare providers may have to engage in this work. 

Ultimately, the request was to explain why one might refer to an ND.  Hopefully all healthcare providers see the value in optimizing health.  MDs, RNs, DOs have their hands full navigating illness, and do so very well.  NDs are well trained to promote health, and are well-positioned to fill that potential gap.